# Synovial Fluid Interleukin Levels Cannot Distinguish between Prosthetic Joint Infection and Active Rheumatoid Arthritis after Hip or Knee Arthroplasty

**DOI:** 10.3390/diagnostics12051196

**Published:** 2022-05-11

**Authors:** Leilei Qin, Chengcheng Du, Jianye Yang, Hai Wang, Xudong Su, Li Wei, Chen Zhao, Cheng Chen, Hong Chen, Ning Hu, Wei Huang

**Affiliations:** 1Department of Orthopaedics, The First Affiliated Hospital of Chongqing Medical University, Chongqing 400016, China; qinleilei@stu.cqmu.edu.cn (L.Q.); duchengcheng@stu.cqmu.edu.cn (C.D.); yangjianye@stu.cqmu.edu.cn (J.Y.); 2018020101@stu.cqmu.edu.cn (H.W.); 2021110216@stu.cqmu.edu.cn (X.S.); 2021120206@stu.cqmu.edu.cn (L.W.); zhaochen@hospital.cqmu.edu.cn (C.Z.); chencheng@hospital.cqmu.edu.cn (C.C.); chenhong@hospital.cqmu.edu.cn (H.C.); 2Orthopedic Laboratory, Chongqing Medical University, Chongqing 400016, China

**Keywords:** prosthetic joint infection, synovial fluid, inflammatory marker, interleukins, rheumatoid arthritis

## Abstract

Inflammatory arthritis affects the level of synovial inflammatory factors, which makes it more difficult to diagnose prosthetic joint infection (PJI) patients with inflammatory arthritis. The aim of this study was to analyze synovial interleukin levels to distinguish between PJI and active rheumatoid arthritis (RA) after a hip or knee arthroplasty. From September 2019 to September 2021, we prospectively enrolled patients with joint pain after arthroplasty due to aseptic prosthesis loosening (n = 39), acute RA (n = 26), and PJI (n = 37). Synovial fluid from the affected joint is obtained and tested with a standard enzyme-linked immunosorbent assay. Receiver operating characteristic curve (ROC) was analyzed for each biomarker. Interleukin (IL)-1β, IL-6, and IL-8 showed promising value in differentiating of aseptic loosening from PJI, with areas under the curves (AUCs) of 0.9590, 0.9506, and 0.9616, respectively. Synovial IL-1β, IL-6, and IL-8 showed limited value in distinguishing between PJI and acute episodes of RA after arthroplasty, with AUCs of 0.7507, 0.7069, and 0.7034, respectively. Interleukins showed satisfactory efficacy in differentiating aseptic loosening from PJI. However, when pain after arthroplasty results from an acute episode of RA, current synovial interleukin levels do not accurately rule out the presence of PJI.

## 1. Introduction

Periprosthetic joint infection (PJI) remains one of the most disastrous complications of arthroplasty, leads to a lengthy hospital stay, creates a substantial economic burden, and increases mortality rates [1,2]. With the in-depth research on the application of novel molecular markers, significant breakthroughs have been made in the diagnosis of PJI, and the whole society has been organized to formulate diagnostic guidelines [3,4]. As far as we know, the distinction between aseptic loosening and PJI, after arthroplasty, is the focus of many current studies due to similar clinical manifestations [5]. It is worth mentioning that the diagnostic value of inflammatory markers in synovial fluid, including the interleukin (IL) family, has been confirmed and validated by numerous studies [6,7,8].

However, with the increasing number of patients undergoing arthroplasty for rheumatoid arthritis (RA), we have to consider how to distinguish between local pain caused by an acute episode of RA or infection after joint replacement [9]. Rheumatoid arthritis interferes with the expression of inflammatory markers, especially during acute episodes [10]. Unfortunately, there is still a lack of evidence for changes in inflammatory markers in synovial fluid after arthroplasty in RA. Studies have shown that arthroplasty is effective in improving joint pain and dysfunction due to RA, and postoperative satisfaction does not differ significantly from that of patients undergoing arthroplasty for osteoarthritis [9]. However, the difference is that RA patients still require regular use of anti-rheumatic drugs after arthroplasty and may still develop joint pain due to an acute episode of RA [11]. Therefore, it is important to distinguish the cause of joint pain after arthroplasty from aseptic loosening, infection of the prosthesis, or acute onset of RA for subsequent treatment.

In this study, synovial fluid was obtained from patients with aseptic loosening and PJI after arthroplasty and from patients with active RA after arthroplasty. We describe the expression of eight interleukins commonly used to identify infectious diseases (including IL-1β, IL-2, IL-4, IL-6, IL-8, IL-10, IL-12, and IL-17) in the synovial fluid of these diseases and explore the value of these interleukins in distinguishing PJI from active RA.

## 2. Materials and Methods

This prospective study was approved by the institutional review board of the authors’ institution, and was registered in the Chinese Clinical Trial Registry, (registration number: ChiCTR1800020440). From September 2019 to September 2021, we prospectively enrolled patients who underwent revision joint arthroplasty due to aseptic prosthesis loosening or PJI after arthroplasty, and patients who experienced pain after joint arthroplasty due to active episodes of RA in rheumatology departments and outpatient clinics. Active RA was defined as Disease Activity Score in 28 joints (DAS28) > 3.2 [12]. Aseptic revision is defined as a single-stage revision surgery caused by loosening, wear, instability, dislocation, etc., except for infection [13]. The definition of PJI was based on the Musculoskeletal Infection Society (MSIS) criteria [14].

Patients with any type of skin ulcer, hematoma, history of recent trauma, or dislocation (within 2 weeks), visible ecchymosis, artificial heart valve, or history of high coagulopathy were excluded. In addition, patients undergoing revision surgery who underwent primary total arthroplasty due to inflammatory arthritis were excluded. Patients enrolled in this study were grouped as follows: patients who underwent revision arthroplasty due to aseptic failure (group A), patients who received antirheumatic therapy after primary arthroplasty due to active RA (group B), and patients who underwent prosthetic removal and spacer implantation for the treatment of PJI (group C).

We recorded patient sex and age, the involved joint and BMI. For synovial fluid collection, all patients underwent joint aspiration immediately after admission to obtain at least 1 mL synovial fluid volume. Within two hours of obtaining the synovial fluid, the specimen was sent to the molecular testing center for retention and testing. All of the synovial samples were tested with a standard enzyme-linked immunosorbent assay (ELISA, R&D Systems). The synovial fluid underwent quantitative analysis of eight common interleukins, including IL-1β, IL-2, IL-4, IL-6, IL-8, IL-10, IL-12, and IL-17. During the revision arthroplasty, at least three tissue samples were collected from the patients for microbiological culture. Serum and synovial fluid samples were tested by the Department of Biotechnology Platform of Laboratory and the Center for Clinical Molecular Medical Detection of Chongqing.

### Statistical Analysis

Statistical analyses were performed by using GraphPad Prism vs. 9.00 for Windows (GraphPad Software, San Diego, CA, USA). Categorical variables were analyzed using chi-squared tests. Continuous variables were analyzed using Mann–Whitney. The non-parametric analyses were performed using the Mann–Whitney U test. Receiver-Operator-Characteristic (ROC) curves were used to assess discriminatory strength of interleukin in synovial fluid between PJI and active RA on the basis of area under the curve (AUC) and to determine optimal cut-off. Sensitivity and specificity for individual values and combinations were calculated. A *p*-value less than 0.05 was considered statistically significant.

## 3. Results

A total of 102 patients were enrolled, of whom 39 underwent aseptic revision, and 37 patients were diagnosed with PJI and underwent stage I prosthesis removal and bone cement filling. Another 26 patients were diagnosed with active RA and were hospitalized in the rheumatology department (Table 1). As can be seen, compared with group A and C, patients in group B were younger (average age was 55.8 + 4.763 years old), and the majority of patients were female (Table 1).

Results of these synovial fluid markers measurements are shown in Figure 1. It can be seen that, unlike aseptic loosening patients, synovial IL-1β, IL-2, IL-6, IL-8, IL-10, and IL-17 were significantly elevated in PJI and active RA groups (Figure 1). However, two other synovial markers, IL-4 and IL-12, were not statistically different between the three groups. Interestingly, synovial IL-1β, IL-8, and IL-10 were significantly higher in PJI patients than in the active RA group (Figure 1). ESR and CRP were also significantly increased in the C group (Appendix A).

Based on these results, we further analyzed six interleukins (IL-1β, IL-2, IL-6, IL-8, IL-10, and IL-17) with significant differences between aseptic loosening group and PJI group (Figure 1). We calculated the optimal cut-off values, sensitivity, and specificity for each of the six interleukins in differentiating aseptic loosening from PJI, and plotted their ROC curves, respectively. The sensitivity and specificity of ESR and CRP in distinguishing aseptic loosening from PJI were also included (Appendix A). As shown in Table 2 and Figure 2, except for IL-17, the other five interleukins have an AUC area of more than 0.85. IL-1β, IL-6, and IL-8 have shown promising value in differentiating the sensitivity and specificity of aseptic loosening from PJI. When the level of synovial IL-1β was 71.03 pg/mL, the AUC area reached 0.9590 (0.9184 to 0.9995), and the sensitivity and specificity were 94.59 (82.30% to 99.04%) and 86.21 (69.44% to 94.50%), respectively. Similarly, when the level of synovial IL-6 was 1327 pg/mL, the AUC area reached 0.9506 (0.9009 to 1.000), and the diagnostic sensitivity and specificity were 90.00 (74.38% to 96.54%) and 89.29 (72.80% to 96.29%), respectively.

Subsequently, we compared three interleukins with significant differences between active RA and PJI and plotted their ROC curves (Figure 3). As can be seen, IL-1β, IL-8, and IL-10 showed AUC values of 0.7507, 0.7069, and 0.7034 in distinguishing RA and PJI, respectively.

## 4. Discussion

Rheumatoid arthritis (RA) is an autoimmune and inflammatory disease characterized by irreversible joint damage and dysfunction [15]. Although the wide utilization of disease-modifying antirheumatic drugs (DMARDs) and biologic agents has improved the quality of life for patients with RA, arthroplasty remains a critical approach to improve activity function and relieving pain in patients with RA who ultimately get severely destroyed joints [16,17]. Total knee arthroplasty (TKA) or total hip arthroplasty (THA) have been shown to significantly improve pain and lower limb function in patients with RA, but patients with RA have an increased incidence of postoperative infection and readmission when compared with patients with osteoarthritis [18,19]. The literature review also revealed that the risk of postoperative infection complications, after total joint arthroplasty (TJA), was increased in patients with RA nearly 2-fold, and deep infection complications increased by 1.5-fold [20].

It is important to note that the diagnosis of chronic atypical infection in deep tissue, after TJA, is challenging compared to acute and symptomatic superficial infection. The manifestations of this chronic infection are mostly localized resting pain and are not significantly different from the inflammatory pain caused by the acute onset of RA. Therefore, distinguishing the cause of joint pain after TJA from infection of aseptic loose prosthesis or acute onset of RA is a topic that must be explored, as it is directly related to further treatment decisions.

Interleukins (IL), as with other cytokines, are not stored in cells and are proteins produced in response to pathogens and other antigens that regulate and mediate inflammatory and immune responses [21]. Based on this property, interleukin has been widely used in the diagnosis of infectious diseases and the evaluation of disease progression [8]. Not surprisingly, interleukin has achieved satisfactory results in the diagnosis of PJI. Elgeidi A et al. found that serum IL-6 was more accurate than conventional markers of inflammation in distinguishing aseptic loosening and PJI after TJA [22]. In subsequent studies, inflammatory markers from synovial fluid were superior to serum sources in diagnosing PJI [23]. In a retrospective clinical study, the synovial interleukin family ((IL-1β), IL-2, IL-4, IL-5, IL-6, IL-8, IL-10, IL-12P70, IL-13, IL-17A, IL-23) was able to accurately diagnose PJI [8,13]. Based on these findings, we attempted to analyze the role of eight synovial interleukins (IL-1β, IL-2, IL-4, IL-6, IL-8, IL-10, IL-12, and IL-17) in post-arthroplasty pain caused by various causes, including aseptic loosening, PJI, and acute onset of RA. Similar to previous studies, six interleukins, including IL-1β, IL-2, IL-6, IL-8, IL-10 and IL-17, were significantly elevated in PJI [8,24,25]. However, not completely similar to previous studies, IL-2 did not show satisfactory diagnostic effect of PJI in Katyayini Sharma’s study [8]. The reason may be that the time interval between specimen collection and detection is too long. Previous studies have also confirmed that the detection accuracy of cytokines is higher when the detection is completed in 4–6 h [26,27]. Among them, IL-1β, IL-2, IL-6, IL-8, and IL-10 have more than 85% AUC value in distinguishing PJI from aseptic loosening. However, only three interleukins (IL-1β, IL-8, and IL-10) showed significant differences between PJI and active RA. Further analysis showed that these three interleukins did not distinguish PJI from active RA well, as AUC values were all below 80% (Figure 3). This suggests that even synovial interleukin is significantly elevated during an acute episode of RA, and that the level of this sterile inflammation is no lower than that of infectious inflammation caused by pathogens.

Arthroplasty is an effective treatment for physical disability caused by RA [16]. However, irregular antirheumatic therapy can still cause synovial inflammation at the surgical site. Synovial proliferation is a potential risk factor for inflammatory episodes because synovial cells are major contributors to the secretion of inflammatory cytokines [28,29]. Thus, synovial interleukin levels are secondary during active RA. Our previous study found significant elevation of synovial IL-6 in patients with active RA, but synovial interleukin expression during an acute episode of RA after arthroplasty was unknown [13]. This study is the first to explore the expression of synovial interleukin in patients undergoing arthroplasty with surgical site pain due to an acute episode of RA. Synovial IL-1β, IL-2, IL-6, IL-8, IL-10, and IL-17 were significantly elevated in patients with active RA compared with patients with aseptic loosening after arthroplasty. Synovial IL-1β, IL-2, IL-6, IL-8, IL-10, and IL-17 were significantly elevated in patients with active RA compared with patients with aseptic loosening after arthroplasty. Furthermore, synovial interleukin levels do not differentiate PJI from active RA due to the high degree of inflammation. This demonstrates the limitations of the eight synovial interleukins present in the diagnosis of PJI.

There are several limitations to this study. First, there is no gold standard test or clinical evaluation for PJI in the literature. We used the Musculoskeletal Infection Society (MSIS) criteria to define the groups as infected, aseptic or active RA. This may lead to the inaccuracy of the existing subgroups, but we believe that the existing subgroups are reliable in combination with the clinical symptoms and etiological results of the patients. The second is the limitation of the small sample size. Despite the small number of patients, the differences in results suggest the limited value of the current eight interleukins in the diagnosis of PJI. Thirdly, only eight interleukins commonly used in our center were explored in this study, which does not mean that all interleukins are not able to effectively judge whether the pain after arthroplasty is caused by acute onset of RA rather than PJI.

In conclusion, among the various interleukins involved in this study, IL-1β, IL-2, IL-6, IL-8, IL-10, and IL-17 showed satisfactory efficacy in differentiating aseptic loosening from PJI. However, when pain after arthroplasty results from an acute episode of RA, current synovial interleukin levels do not accurately rule out the presence of PJI. Therefore, when autoimmune diseases are present, more direct evidence is needed to determine what is responsible for pain after arthroplasty.

## Figures and Tables

**Figure 1 diagnostics-12-01196-f001:**
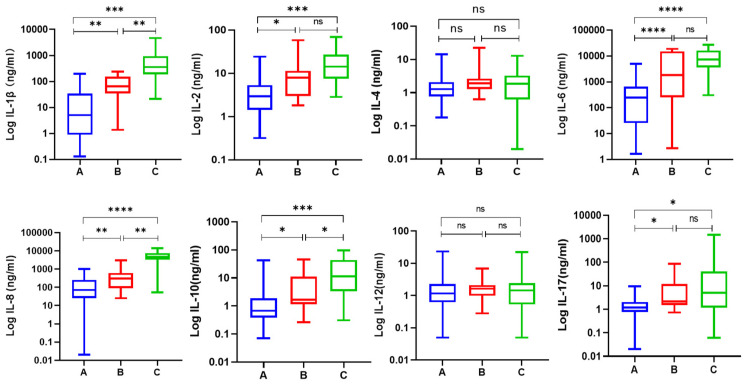
Scatterplots showing the concentration of each biomarker in the three groups. ns, not significant. * *p*-value < 0.05; ** *p*-value < 0.01; *** *p*-value < 0.001; **** *p*-value < 0.0001.

**Figure 2 diagnostics-12-01196-f002:**
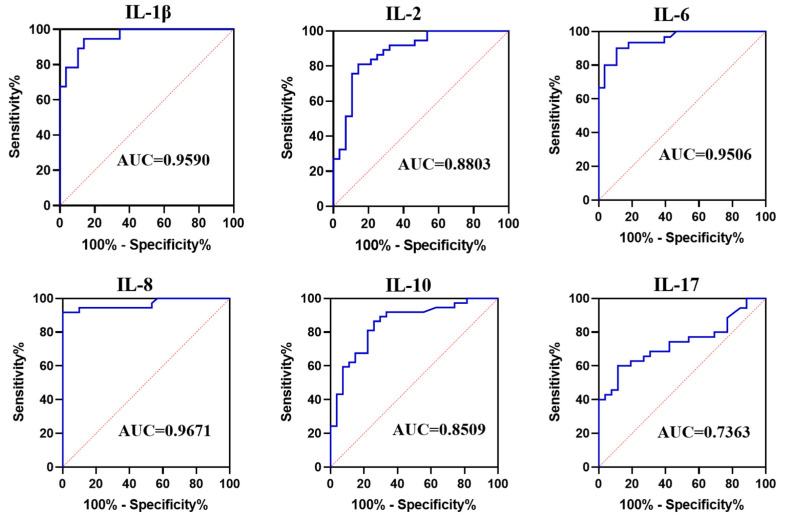
Receiver Operating Characteristic curves (ROCs) and area under curves (AUC) of partial synovial fluid interleukin in distinguishing between PJI and aseptic loosening.

**Figure 3 diagnostics-12-01196-f003:**
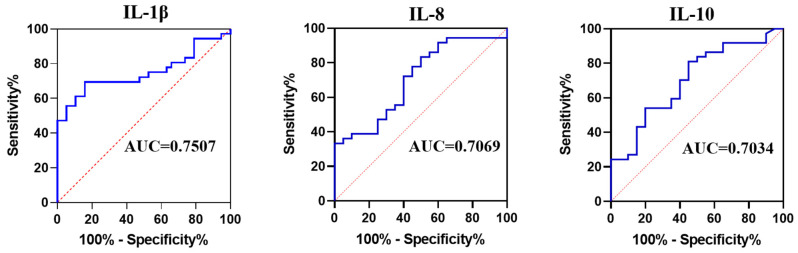
Receiver Operating Characteristic curves (ROCs) and area under curves (AUC) of partial synovial fluid interleukin in distinguishing between PJI and active RA.

**Table 1 diagnostics-12-01196-t001:** Demographic data for the study population.

Variable	Group A (N = 39)	Group B (N = 26)	Group C (N = 37)	P1-Value	P2-Value
Sex				0.9999 *	0.0003 *
Male	17 (43.6%)	3 (12.5%)	16 (43.2%)		Male
Female	22 (56.4%)	23 (88.5%)	21 (56.8%)		
Joint type				0.2509 *	<0.0001 *
Knee	16 (41.03%)	26 (100%)	21 (56.8%)		
Hip	23 (58.97%)	NA	16 (43.2%)		
Age, (yr)	62.0 ± 8.386	55.8 ± 4.763	64.9 ± 6.817	0.1300 #	0.0047 #
BMI, (kg/m^2^)	24.04 ± 3.350	22.68 ± 2.278	23.65 ± 3.044	0.6041 #	0.0998 #
Comorbidities (n)					
Hypertension	15	7	21		
Diabetes	8	5	17		
Cardiovascular disease	2	0	4		

P1: Between Group A and C, P2: Between Group B and C, Variables are expressed as mean ± SD, or numbers (percentage), BMI, body mass index, * Chi squared test, # Mann–Whitney U test.

**Table 2 diagnostics-12-01196-t002:** Sensitivity and Specificity of inflammatory markers.

Markers	Cut-Off Value (pg/mL)	AUC (95% CI)	Sensitivity (95% CI)	Specificity (95% CI)
IL-1β	71.03	0.9590 (0.9184 to 0.9995)	94.59 (82.30% to 99.04%)	86.21 (69.44% to 94.50%)
IL-2	6.50	0.8803 (0.7945 to 0.9661)	81.08 (65.80% to 90.52%)	85.71 (68.51% to 94.30%)
IL-6	1327	0.9506 (0.9009 to 1.000)	90.00 (74.38% to 96.54%)	89.29 (72.80% to 96.29%)
IL-8	1033	0.9616 (0.9172 to 1.000)	86.11 (71.34% to 93.92%)	100.0 (88.65% to 100.0%)
IL-10	1.48	0.8509 (0.7553 to 0.9464)	86.49 (72.02% to 94.09%)	74.07 (55.32% to 86.83%)
IL-17	2.95	0.7363 (0.6117 to 0.8609)	60.00 (43.57% to 74.45%)	88.46 (71.02% to 96.00%)

CI, confidence interval.

## Data Availability

Not applicable.

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
