# Peer review of "Synovial Fluid Interleukin Levels Cannot Distinguish between Prosthetic Joint Infection and Active Rheumatoid Arthritis after Hip or Knee Arthroplasty"

_diagnostics, 2022, doi:10.3390/diagnostics12051196_

Round 1

Reviewer 1 Report

This is an interesting study on the role of synovial interleukins in differentiating acute joint paint due to either aseptic loosening, periprosthetic joint infection or rheumatoid arthritis flare following total hip or knee arthroplasty. Overall, the manuscript is well written and the methodology partially supports the results. I believe that major changes should be made before considering publication.

Some major comments:

  • The clinical benefit of investigating interleukin synovial concentration in addition to MSIS criteria in the subgroups considered is not clear. More specifically, authors should deeply discuss what are the grey zones in which MSIS criteria may miss or misdiagnose a PJI in patients with aseptic loosening or arthritic joint pain.
  • According to MSIS criteria, were synovial WBC count or LE, alpha defensin, PMN or CRP evaluated in all groups? It would be interesting to correlate such parameters with the interleukins investigated (e.g., the coexistence of high alpha-defensin and IL-1b levels are more predictive of PJI etc.)
  • What about RA patients with joint pain due to aseptic loosening? Were they included in group A or B, or even excluded?
  • Additional demographic information should be added to Table 1, including comorbidities and mean time to joint replacement.

Some minor comments:

  • Title: PJI and RA should be replaced with their extended form. The "a" article should be removed before "arthroplasty".
  • According to Author's Guidelines, the abstract should be unstructured. Please remove "Background", "Methods", "Results" and "Conclusions".
  • Please be consistent in the use of abbreviated forms and always put corresponding extended forms before (e.g., RA is used for the first time in line 13 without any extended form before).
  • Extensive English language polishing is warmly advised, with particular attention to verb tenses (please prefer past tenses), syntax and use of articles.
  • Lines 80-81: please mention the manufacturer of ELISA reagents.
  • Table 1: the corresponding percentage of knees in group B is missing.

Author Response

April 20, 2022                  

                                          Response to comments of diagnostics-1694582

Synovial fluid interleukin levels cannot distinguish between prosthetic joint infection and active rheumatoid arthritis after hip or knee arthroplasty

Thank you very much for your excellent and insightful comments and suggestions on our manuscript entitled “Synovial fluid interleukin levels cannot distinguish between prosthetic joint infection and active rheumatoid arthritis after hip or knee arthroplasty” (diagnostics-1694582). We have carefully followed the suggestions of the referee and modified the manuscript. We hope that the editor and reviewer will find our response to their queries satisfied and the revised manuscript is ready for publication. Please see below for details:

Response to the Reviewer #1

1-     The clinical benefit of investigating interleukin synovial concentration in addition to MSIS criteria in the subgroups considered is not clear. More specifically, authors should deeply discuss what are the grey zones in which MSIS criteria may miss or misdiagnose a PJI in patients with aseptic loosening or arthritic joint pain.  

We appreciate the reviewer’s comment. The current MSIS criteria are of great value for the diagnosis of PJI and are still being improved. The deficiency is that the cases involved did not involve patients with inflammatory arthritis. Therefore, it is of great significance to explore the expression of interleukin in patients with RA.

2-     According to MSIS criteria, were synovial WBC count or LE, alpha defensin, PMN or CRP evaluated in all groups? It would be interesting to correlate such parameters with the interleukins investigated (e.g., the coexistence of high alpha-defensin and IL-1b levels are more predictive of PJI etc.)

We appreciate the reviewer’s insightful suggestions. The LE and Alpha defensin tests specified in the MSIS standard make sense, but unfortunately our agency does not have the capability to do so. In the active RA group, synovial fluid testing was limited to interleukin and bacterial cultures. We strongly agree with the suggestions of reviewers. Meanwhile, we uploaded ESR/CRP and other indicators as supplementary data. We found that ESR/CRP had no significant difference between active RA and PJI, and could only be used as an inflammatory marker to distinguish PJI from aseptic loosening.

3-    What about RA patients with joint pain due to aseptic loosening? Were they included in group A or B, or even excluded?

  We identify with this critical issue of reviewers. In our cohort, RA patients from the rheumatology department experienced pain relief after antirheumatic therapy; none of these patients underwent revision surgery. In contrast, aseptic loosening and PJI patients do not have a history of RA. However, the suggestion of reviewers is a very meaningful direction of inquiry.

4-    Additional demographic information should be added to Table 1, including comorbidities and mean time to joint replacement.

  Thanks to reviewers for their suggestions. Common medical complications in each group were added in Table 1. As the primary arthroplasty of some patients is not completed in our institution, it is temporarily impossible to accurately calculate the operation time.

5-    Title: PJI and RA should be replaced with their extended form. The "a" article should be removed before "arthroplasty".

We strongly agree with the suggestions of reviewers, and have corrected the title to " Synovial fluid interleukin levels cannot distinguish between prosthetic joint infection and active rheumatoid arthritis after hip or knee arthroplasty".

6-    According to Author's Guidelines, the abstract should be unstructured. Please remove "Background", "Methods", "Results" and "Conclusions".

  According to the suggestions of reviewers, we have changed the abstract in the manuscript as required.

7-    Please be consistent in the use of abbreviated forms and always put corresponding extended forms before (e.g., RA is used for the first time in line 13 without any extended form before).

  Thanks for the reviewer's reminder, we have corrected it in the manuscript.

8-    Extensive English language polishing is warmly advised, with particular attention to verb tenses (please prefer past tenses), syntax and use of articles.

According to the comments of reviewers, we carefully examined the language structure of the manuscript and made linguistic polishing.

9-    Lines 80-81: please mention the manufacturer of ELISA reagents.

Thanks to the reviewer's suggestion, we have added the ELISA reagent manufacturer to the manuscript.

10-   Table 1: the corresponding percentage of knees in group B is missing.

According to the comments of reviewers, we have added the percentage of knees in group B to Table 1.

Sincerely yours,

Wei Huang

Wei Huang, PhD, Professor, and Director

 Department of Orthopaedics

Chongqing Medical University First Affiliated Hospital

Chongqing, 400016, People’s Republic of China

Phone: +86 138 8338 3330

E-mail: huangwei68@263.net

Reviewer 2 Report

Interesting article with analysis of PJI. The paper is well prepared, there are no methodological mistakes, the limitations of the study are well described. Additionally, interesting would be to see the curves of specificity/sensitivity with multivariate marker combined (doubles or triples:  IL-1b, IL-6, IL-8) 

Author Response

April 20, 2022                  

                                          Response to comments of diagnostics-1694582

Synovial fluid interleukin levels cannot distinguish between prosthetic joint infection and active rheumatoid arthritis after hip or knee arthroplasty

Thank you very much for your excellent and insightful comments and suggestions on our manuscript entitled “Synovial fluid interleukin levels cannot distinguish between prosthetic joint infection and active rheumatoid arthritis after hip or knee arthroplasty” (diagnostics-1694582). We have carefully followed the suggestions of the referee and modified the manuscript. We hope that the editor and reviewer will find our response to their queries satisfied and the revised manuscript is ready for publication. Please see below for details:

Response to the Reviewer #2

Interesting article with analysis of PJI. The paper is well prepared, there are no methodological mistakes, the limitations of the study are well described. Additionally, interesting would be to see the curves of specificity/sensitivity with multivariate marker combined (doubles or triples:  IL-1b, IL-6, IL-8)

Thanks for the reviewer's recognition of our study, we will continue to focus on the research related to periprosthetic infection after arthroplasty.

Sincerely yours,

Wei Huang

Wei Huang, PhD, Professor, and Director

 Department of Orthopaedics

Chongqing Medical University First Affiliated Hospital

Chongqing, 400016, People’s Republic of China

Phone: +86 138 8338 3330

E-mail: huangwei68@263.net

Reviewer 3 Report

Thank you for giving me the opportunity to review this valuable manuscript. The study takes in account the diagnostic value of inflammatory markers, specifically interleukins, in synovial fluid, confirming the possible use of some of them in discriminating between aseptic loosening and prosthetic joint infection (PJI). Furthermore authors try using the same interleukins to discriminate between acute episode of AR and PJI, an interesting and useful challenge.  

The work seems well designed, well written and well argued. Although it has some limitations, as admitted by the authors, mostly the small sample size, it remains interesting and valuable.

Just some notes:

I would suggest including, if available, generic serological inflammation markers such as CRP / ESR / WBC, to better frame the patients and to add value to the study.

Strangely enough, the concentration values of some interleukins are completely different from those in other works cited: ng versus pg??? Have they been correctly reported in figure 1?

Another interesting fact is that IL-2 level was different between aseptic loosening and PJI while in other works (8 = Katyayini Sharma) was not statistical different, maybe in discussion could be taken in account…

M&M: It would be more correct to have some more specifications about the ELISA kits used, or at least the detection range.

I would recommend keeping the same sequence of groups A B C... through all sections of the work, for a more immediate understanding:

Abstract: aseptic prosthesis loosening (n=39), // PJI (N=37), // and acute RA (n=26)

Intro: aseptic loosening // periprosthetic infection //with active RA

M&M: group A:  aseptic failure // group B: active RA // group C: PJI

Results: aseptic revision // PJI // active RA

I mean that groups are often listed not in alphabetical sequence... A-C-B

Typos

(Figuer III).  Line 122

Author Response

April 20, 2022                  

                                          Response to comments of diagnostics-1694582

Synovial fluid interleukin levels cannot distinguish between prosthetic joint infection and active rheumatoid arthritis after hip or knee arthroplasty

Thank you very much for your excellent and insightful comments and suggestions on our manuscript entitled “Synovial fluid interleukin levels cannot distinguish between prosthetic joint infection and active rheumatoid arthritis after hip or knee arthroplasty” (diagnostics-1694582). We have carefully followed the suggestions of the referee and modified the manuscript. We hope that the editor and reviewer will find our response to their queries satisfied and the revised manuscript is ready for publication. Please see below for details:

Response to the Reviewer #3

  • I would suggest including, if available, generic serological inflammation markers such as CRP / ESR / WBC, to better frame the patients and to add value to the study.

Thanks to reviewers for their valuable suggestions. We uploaded ESR/CRP and other indicators as supplementary data, and found that ESR/CRP showed no significant difference between active RA and PJI, and could only be used as an inflammatory marker to distinguish PJI from aseptic loosening.

  • Strangely enough, the concentration values of some interleukins are completely different from those in other works cited: ng versus pg??? Have they been correctly reported in figure 1?

Thank you for your questions. In Figure 1, we adopted the concentration unit of markers as ng instead of pg, because the levels of markers in the three groups were too different. The unified unit of ng made the visualization better, but their values were all correctly displayed.

  • Another interesting fact is that IL-2 level was different between aseptic loosening and PJI while in other works (8 = Katyayini Sharma) was not statistical different, maybe in discussion could be taken in account…

Thanks to reviewers for their valuable suggestions. In our study, IL-2 showed outstanding value in distinguishing aseptic loosening from PJI, which is different from the findings of Katyayini Sharma et al. The reason may be that the time interval between specimen collection and detection is too long. Previous studies have also confirmed that the detection accuracy of cytokines is higher when the detection is completed in 4-6h. We further add relevant discussions to the discussion section of the manuscript (pp. 179-184).

  • M&M: It would be more correct to have some more specifications about the ELISA kits used, or at least the detection range.

Thanks to the reviewer's suggestion, we have added the ELISA reagent manufacturer to the manuscript.

  • I would recommend keeping the same sequence of groups A B C... through all sections of the work, for a more immediate understanding:

Abstract: aseptic prosthesis loosening (n=39), // PJI (N=37), // and acute RA (n=26)

Intro: aseptic loosening // periprosthetic infection //with active RA

M&M: group A:  aseptic failure // group B: active RA // group C: PJI

Results: aseptic revision // PJI // active RA

I mean that groups are often listed not in alphabetical sequence... A-C-B

We appreciate the reviewer's comments. We checked the order of expression of the group and made adjustments in the manuscript.

  • Typos

(Figuer III).  Line 122  

We thank the reviewers for their corrections and have made changes in the manuscript.

Sincerely yours,

Wei Huang

Wei Huang, PhD, Professor, and Director

 Department of Orthopaedics

Chongqing Medical University First Affiliated Hospital

Chongqing, 400016, People’s Republic of China

Phone: +86 138 8338 3330

E-mail: huangwei68@263.net

Round 2

Reviewer 1 Report

The authors have addressed most reviewers' comments and made notable improvements to their manuscript.

Some additional minor points:

  • A space should be inserted between the text and the reference numbers whenever applicable.
  • The abbreviated form for "interleukin" should be mentioned in line 37.
  • Please be consistent with the use of abbreviations after mentioning them the first time (e.g., use PJI instead of periprosthetic joint infection in line 52).

Author Response

April 26, 2022                  

                                          Response to comments of diagnostics-1694582

Thank you very much for your excellent and insightful comments and suggestions on our manuscript entitled “Synovial fluid interleukin levels cannot distinguish between prosthetic joint infection and active rheumatoid arthritis after hip or knee arthroplasty” (diagnostics-1694582). We have carefully followed the suggestions of the referee and modified the manuscript. We hope that the editor and reviewer will find our response to their queries satisfied and the revised manuscript is ready for publication. Please see below for details:

Response to the Reviewer #1

1-     A space should be inserted between the text and the reference numbers whenever applicable.

Thanks to the reviewer's suggestion, we added a space between the text and the reference number in the manuscript.

2-     The abbreviated form for "interleukin" should be mentioned in line 37.

Please be consistent with the use of abbreviations after mentioning them the first time (e.g., use PJI instead of periprosthetic joint infection in line 52).

  Thanks to the reviewer's reminder, we checked the manuscript and corrected the abbreviation of interleukin to the specified position. Meanwhile, we have replaced the periprosthetic joint infection with PJI.

Sincerely yours,

Wei Huang

Wei Huang, PhD, Professor, and Director

 Department of Orthopaedics

Chongqing Medical University First Affiliated Hospital

Chongqing, 400016, People’s Republic of China

Phone: +86 138 8338 3330

E-mail: huangwei@hospital.cqmu.edu.cn
